# Addictive use of digital devices in young children: Associations with delay discounting, self-control and academic performance

**Tim Schulz van Endert** *

School of Business and Economics, Freie Universität, Berlin, Germany

* tim.vanendert@fu-berlin.de

## Abstract

The use of smartphones, tablets and laptops/PCs has become ingrained in adults' and increasingly in children's lives, which has sparked a debate about the risk of addiction to digital devices. Previous research has linked specific use of digital devices (e.g. online gaming, smartphone screen time) with impulsive behavior in the context of intertemporal choice among adolescents and adults. However, not much is known about children's addictive behavior towards digital devices and its relationship to personality factors and academic performance. This study investigated the associations between addictive use of digital devices, self-reported usage duration, delay discounting, self-control and academic success in children aged 10 to 13. Addictive use of digital devices was positively related to delay discounting, but self-control confounded the relationship between the two variables. Furthermore, self-control and self-reported usage duration but not the degree of addictive use predicted the most recent grade average. These findings indicate that children's problematic behavior towards digital devices compares to other maladaptive behaviors (e.g. substance abuse, pathological gambling) in terms of impulsive choice and point towards the key role self-control seems to play in lowering a potential risk of digital addiction.

## Introduction

Digital devices, such as smartphones, tablets and laptops, have become an integral part in the lives of the majority of people around the world. Recent surveys e.g. in the US estimate that 81% of adults own a smartphone, 74% own a laptop and 52% own a tablet [1]. Notably, not only adults but also children have been increasingly surrounded by digital devices; a report from the UK states that in 2019 more than two thirds of 5- to 16-year-olds owned a smartphone and that 80% of 7- to 16-year-olds had internet access in their own room [2]. The same report also estimates that children's average time spent online is 3.4 hours per day, with the main activities being watching videos (e.g. on YouTube and TikTok), using social media (e.g. Instagram and Snapchat) or gaming (e.g. Fortnite or Minecraft). These numbers have seen an unprecedented increase since the COVID-19 pandemic, which has, to a large extent, forced children to remain home, receive online schooling and interact with friends digitally. While the effects of these measures vary from country to country, a 163% increase in daily screen

**Data Availability Statement:** All relevant data are within the manuscript and its Supporting Information files.

**Funding:** The author received no specific funding for this work.

**Competing interests:** The authors have declared that no competing interests exist.

time during the first lockdown in Germany is not an unusual occurrence as observed by Schmidt et al. [3].

These developments have added momentum to the debate about the addiction potential of digital devices—especially for children, who are particularly at risk of developing addictive behaviors [4]. Evidence of negative implications of excessive digital device use, such as stress [5], sleep disturbance [6] or poor academic performance [7], has accumulated in recent years. However, researchers have not yet agreed on a standardized definition of digital addiction, which clearly separates it from other, possibly underlying disorders [8]. For one aspect of problematic use of digital devices, namely Internet Gaming Disorder, existing research has matured to stage where it suggests a potential future inclusion in the Diagnostic and Statistical Manual of Mental Disorders, Fifth Edition (DSM-5), as an officially diagnosable condition. Other aspects, such as smartphone addiction, are less mature and the literature has so far only identified a significant overlap between addiction to smartphones and substance-related disorders defined in the DSM-5 [9, 10]. One area of research, which seeks to explore overall addiction to digital devices, encompassing various media (e.g. smartphones, tablets and laptops/PCs) and activities (e.g. gaming, social media), seems promising but is still in its infancy [11]. Unlike the aforementioned strands of literature, research on overall digital addiction takes into account the newly emerged usage behavior of performing a multitude of activities on and across several different digital devices (e.g. sending WhatsApp messages on a smartphone, playing games on a tablet and watching movies on a laptop). This may promote a degree of consolidation of the large number of concepts of technology addiction and their corresponding scales, which have emerged over the years but have recently been shown to be highly similar on a dimensional level [12].

A scale assessing digital addiction particularly among young children was recently introduced [11]. The Digital Addiction Scale for Children (DASC) measures to which degree children's use of smartphones, tablets and laptops/PCs negatively affects their educational, psychological, social and physical well-being. To account for the ongoing debate about a standardized definition of digital addiction and the corresponding lack of a firm diagnosis, throughout this paper the softer formulation "addictive use of digital devices" is used rather than "digital addiction" when referring to children's use of digital devices with adverse consequences. To further our understanding of this behavioral pattern and enable possible future intervention, the scale needs to be investigated in connection with personality factors, which may contribute to problematic behavior towards digital devices [13].

In this context, delay discounting, i.e. the tendency to discount rewards as a function of the delay of their delivery, suggests itself as an avenue for research. This cognitive process underlies human and non-human animals' preference for smaller, immediate rewards over larger, delayed rewards and is often used as a measure of impulsivity [14]. Delay discounting has been studied extensively in the past decades, mostly by means of intertemporal choice problems, in which participants are faced with the tradeoff between the amount and the delay of a reward (e.g. choosing between 100€ today or 150€ in one month). Several models seeking to capture behavior have emerged, with hyperbolic discounting providing the best fit for most empirical data [15]. Its equation $V = A / (1+kD)$ ($V$ is the present value of the future reward, $A$ is the reward amount and $D$ is the delay to the reward) contains one free parameter $k$, which represents an individual's discount rate. The lower this discounting parameter, the less the individual devalues future rewards and is therefore relatively less impulsive than a person with a higher discount rate. Due to the relative temporal stability of individuals' discount rates, delay discounting may be seen as a trait variable [16]. Also, a plethora of studies has shown an association between delay discounting and a variety of maladaptive behaviors, such as substance abuse [17], smoking [18] and pathological gambling [19, 20] or overeating [21]. In these

studies, addicted individuals discounted future rewards more steeply than control subjects, which makes delay discounting a reliable indicator for various kinds of addictions [22]. Given that the discounting of future rewards is not only related to substance-based but also to behavioral addictions, this raises the question if delay discounting is also associated with addictive use of digital devices. Past studies have only been able to show relationships between delay discounting and single aspects of digital use, such as internet gaming [23] or smartphone screen time [24]. In addition, the samples studied consisted of adolescents or adults, despite regular use of digital devices already starting in childhood [2].

Furthermore, researchers agree on the key role of self-control in the development [25] and treatment [26] of addictive behaviors. On the one hand, a decreased ability to regulate thoughts and emotions contributes to risk-taking behavior, such as initiating use of addictive drugs, which is a common phenomenon in adolescents [27]. On the other hand, impaired self-control is a key symptom of addicted individuals, i.e. the inability to stop engaging in addictive behavior despite a willingness to do so. Thus, behavioral training to strengthen control functions has been proposed as an effective approach to reduce addiction [28]. Additionally, prominent models of decision-making have also highlighted self-control as a mechanism underlying delay discounting [22, 29]. According to these accounts, exertion of self-control suppresses the impulse of choosing a smaller, immediate reward and biases choice behavior towards the larger, delayed reward. However, the interrelationships between delay discounting, addictive use of digital devices and self-control have yet to be explored.

Lastly, a number of studies have shown an association between various kinds of addictive behavior and poor academic performance [30–32]. Being distracted in the classroom or while studying, concentration lapses due to lack of sleep or missing classes and exams have been put forth as explanations for this finding. Given the novelty of the concept of digital addiction, the question whether the pattern suggested by the literature also holds in the context of addictive use of digital devices, particularly by young children, needs empirical investigation. This issue is of great importance as fundamental reading, writing and mathematics skills are taught at this stage. It is also relevant for the debate about increasingly integrating digital media in classroom activities and homework as part of the digitalization of schools. Therefore, this present study examines the following three hypotheses:

**H1:** Delay discounting is positively correlated with children's addictive use of digital devices

**H2:** Self-control is negatively correlated with children's addictive use of digital devices

**H3:** Children's addictive use of digital devices is negatively correlated with academic success

This study contributes to the literature by showing behavioral similarities between addictive use of digital devices and other problematic behaviors, by highlighting the central role that self-control seems to play in the context of digital addiction and by uncovering an intriguing pattern when comparing the relationships of problematic use vs. raw usage duration of digital devices with academic success.

## Methods

### Participants

75 children aged 10 to 13 (mean 11.3 years, 47% female) with no officially diagnosed mental disorders (e.g. Attention-Deficit/Hyperactivity Disorder, Obsessive-Compulsive Disorder) were recruited from a public elementary school in Berlin, Germany. The participants were 5th and 6th grade students and were selected for two reasons. On the one hand, participants needed to be able to understand the tasks and questionnaires employed in this study. On the

other hand, this age represents a major crossroad for the children of Berlin; within the city's school system, students graduate from elementary school after 6th grade and progress to either high school ("Gymnasium") or integrative secondary school ("Integrierte Sekundarschule") depending on their academic performance (a German high school diploma provides eligibility to attend University, while students from Integrative Secondary School graduate after 9th or 10th grade in order to start an apprenticeship). This age group, prior to above-mentioned separation, thus had the positive side effect of implying a variety of academic skills as well as socio-economic backgrounds. Furthermore, the school's headmaster affirmed that there was a significant diversity of ethnicities and nationalities among students and that no mental disorders existed in the observed classes. Parents (or guardians) of all participants were informed that participation was voluntary as well as anonymous and did not have an impact on their children's grades. Roughly 15% of invited students chose not to participate in the study. Informed consent documents were signed before each study session.

## Measures

**Addictive use of digital devices.**   To measure the degree of addictive use of smartphones and tablets the Digital Addiction Scale for Children (DASC) [11] was employed. The DASC is a 25-item self-report instrument based on the theoretical framework of DSM-5 Internet Gaming Disorder as well as on the components model of addiction [33]. The resulting nine addiction criteria are Preoccupation, Tolerance, Withdrawal, Problems, Conflict, Deception, Displacement, Relapse and Mood Modification, each represented by two to four items within the scale. Scores range from 25 to 125, higher scores indicating a greater risk of addiction to digital devices. As only the degree of smartphone and tablet use with adverse consequences rather than the identification of addicts was relevant to this study, the scale was not used to distinguish between addicts and non-addicts in the analyses. Correspondingly, throughout this paper the formulation "addictive use of digital devices" is used rather than "digital addiction", to avoid suggesting a firm diagnosis, which is not available at the moment. The scale was specifically developed for 9- to 12-year-old children and has been shown to be a reliable and valid instrument to assess the risk of being addicted to digital devices [11]. A German translation of the DASC was used, after having been checked for understandability by one 5th grade and one 6th grade teacher independently. The internal consistency of the scale was excellent ($\alpha$ = 0.94).

**Delay discounting.**   The participants' preference for smaller immediate rewards over larger delayed rewards was assessed with a German translation of the 27-item Monetary Choice Questionnaire [17]. In this questionnaire participants repeatedly choose between a smaller, immediately available reward and a larger reward available in the future, all rewards being hypothetical and consisting of small (e.g. €20), medium (e.g. €54) and large amounts of money (e.g. €78). The proportion of choices of the larger delayed reward (LDR) is used as a measure of impulsivity, i.e. the lower the proportion, the more impulsive the individual. The scale is widely used in the literature for studying adults and has also been shown to be a valid instrument for young children [34–36]. Also, the Monetary Choice Questionnaire provides similar results to more extended instruments [37] as well as to paradigms that use real or potentially real rewards [38]. Furthermore, the proportion of LDR measure is a simple yet reliable and valid measure, which does not require the assumption of hyperbolic discounting [39]. Within the present dataset the LDR proportion was highly correlated (r = -0.98, p<0.001) with the natural log of the discount parameter k according to Kirby et al. [17], indicating that the LDR measure was accurately assessing participants' discounting of future rewards. The responses to the Monetary Choice Questionnaire were scored using automated scoring [40]. This tool also provides consistency scores in order to identify insufficient comprehension or a

lack of or attending to the questionnaire. Three participants had consistency scores below 75%, the recommended threshold for good quality of responses [41], resulting in their exclusion from the analyses.

**Self-control.** As a measure of self-control the German adaptation [42] of Tangney et al.'s Brief Self-Control Scale [43] was used, which is a widely used self-report measure of trait self-control. Scores range from 13 to 65, higher scores representing better ability to regulate thoughts, emotions and behavior. Research has shown that the 13-item brief self-control scale provides equally reliable and valid results as the long version [43] and is appropriate for use with young children [44]. Good internal consistency was indicated by Cronbach's α of 0.75.

**Additional variables.** As a measure of academic performance, the most recent semester's grade average with a possible range from 1.0 (best possible, "straight A") to 6.0 (worst possible, "straight F") was used. Also, children were asked to estimate their average daily duration of several popular activities on digital devices (e.g. social media, games) as well as their typical total screen time on weekdays and weekends to attain various measures for self-reported usage, thereby allowing for robustness checks of results. Lastly, age, gender, years of smartphone ownership and weekly pocket money were elicited as control variables.

## Procedure

Data was collected in June 2020, several weeks after reopening of schools following the initial seven-week lockdown in Germany. The study was conducted in five sessions, which were held in the school's computer lab. At the beginning of each session, the researcher instructed participants about the tasks, while a teacher assisted in ensuring a setting comparable to class examination (silence, no copying from neighbors etc.). Throughout all sessions the order of tasks was fixed as follows: 1) self-reported usage patterns, 2) Monetary Choice Questionnaire, 3) Digital Addiction Scale, 4) control variables and 5) Brief Self-Control scale. One session lasted about 30 minutes. The study was approved by the Central Ethics Committee of the Freie Universität Berlin (approval no. 2020–005)

## Results

### Addictive use of digital devices and delay discounting

To investigate the first hypothesis, initially the relationship between scores of the DASC and the LDR proportion was analyzed. A negative correlation (r = -0.28, p = 0.016) between the two variables was found. On average, the more often children chose the larger delayed reward, the less they addictively used digital devices. When breaking down the DASC into its nine subscales, delay discounting was significantly correlated with Withdrawal (r = 0.30, p = 0.014), Deception (r = 0.24, p = 0.043) and Mood Modification (r = 0.29, p = 0.010). Next, to control for possible effects of gender, age, years of ownership of digital devices and pocket money a regression analysis was performed with the control variables and the proportion of LDR choices as independent variables and the overall DASC score as the dependent variable. All assumptions for multiple regression analysis were met. As shown in Table 1, the LDR proportion was the only significant predictor of scores in the DASC (β = -0.27, p = 0.032). The overall model yielded an $R^2$ of 0.10, F-statistic of 1.50 and p-value of 0.201. The similarity in correlation patterns of the natural log of the discount parameter k and the LDR proportion indicated that the latter measure was accurately assessing participants' delay discounting. Additionally, self-reported usage of digital devices was positively correlated to the DASC score (r = 0.37, p = 0.001), but no relationship was found with delay discounting (r = 0.09, p = 0.465). Self-reported usage was positively related to the DASC subscales Preoccupation, Withdrawal, Displacement, Relapse and Problems, the latter showing the strongest correlation of r = 0.41

**Table 1. Multiple regression analysis of predictors of DASC score.**

| Term | B | SE B | 95% CI | | β | t | p |
|---|---|---|---|---|---|---|---|
| | | | LL | UL | | | |
| Intercept | 85.41 | 24.62 | 36.25 | 134.56 | 0.00 | 3.47 | 0.001 |
| Age | -2.26 | 2.25 | -6.74 | 2.22 | -0.12 | -1.01 | 0.318 |
| Gender (Male) | 1.26 | 2.09 | -2.92 | 5.44 | 0.07 | 0.60 | 0.550 |
| Pocket money | -0.06 | 0.19 | -0.43 | 0.31 | -0.04 | -0.32 | 0.748 |
| Years of ownership | 0.21 | 1.62 | -3.03 | 3.44 | 0.02 | 0.13 | 0.899 |
| LDR proportion | -18.08 | 8.23 | -34.51 | -1.65 | -0.27 | -2.20 | 0.032 |

Note: Effect coding was applied for categorical variables.

(p<0.001). Table 2 shows bivariate correlations of the main variables in this study. The breakdown of the DASC and its correlations with key variables can be found in S2 Table.

## Addictive use of digital devices and self-control

The second hypothesis required an investigation of the relationship between addictive digital device use and self-control. There was a strong negative correlation between self-control and the DASC score (r = -0.69, p<0.001). On average, the higher children's scores were on the brief self-control scale the less they tended to addictively use digital devices. Correspondingly, self-control was negatively associated with all nine subscales of the DASC, having the strongest relationship with Tolerance (r = -0.65, p<0.001). Furthermore, self-control was also correlated to the LDR proportion (r = 0.25, p = 0.034), indicating possible confounding between addictive behavior and delay discounting. Therefore, a regression analysis was performed with the control variables (gender, age, years of ownership of digital devices and pocket money), self-reported usage, self-control as well as the proportion of LDR choices as independent variables and the overall DASC score as the dependent variable. As displayed in Table 3, self-control (β = -0.58, p<0.001) and self-reported usage (β = 0.32, p = 0.003) were the only significant predictors of the DASC score. Notably, with the variable self-control in the model the LDR proportion no longer significantly predicted the DASC score (β = -0.15, p = 0.100). The overall model's $R^2$ was 0.59 with an F-statistic of 13.21 and p-value of <0.001.

**Table 2. Correlations between main variables.**

| Variable | 1. | 2. | 3. | 4. | 5. | 6. | 7. | 8. | 9. |
|---|---|---|---|---|---|---|---|---|---|
| 1. DASC score | - | | | | | | | | |
| 2. Self-reported usage | 0.37** | - | | | | | | | |
| 3. LDR proportion | -0.28* | 0.09 | - | | | | | | |
| 4. ln overall k | 0.27* | -0.07 | -0.98*** | - | | | | | |
| 5. Self-control | -0.69*** | -0.20 | 0.25* | -0.23* | - | | | | |
| 6. Grade average | 0.15 | 0.43*** | 0.02 | -0.03 | -0.31* | - | | | |
| 7. Age | -0.16 | 0.09 | 0.12 | -0.11 | 0.06 | 0.21 | - | | |
| 8. Years of ownership | 0.00 | 0.36** | -0.06 | 0.07 | 0.12 | 0.16 | 0.22 | - | |
| 9. Pocket money | 0.00 | 0.43*** | -0.19 | 0.20 | 0.11 | 0.10 | 0.10 | 0.25* | - |

*p < 0.05

**p < 0.01

*** p < 0.001.

**Table 3. Multiple regression analysis of predictors of DASC score.**

| Term | B | SE B | 95% CI | | β | t | p |
|---|---|---|---|---|---|---|---|
| | | | LL | UL | | | |
| Intercept | 123.12 | 18.94 | 85.28 | 160.95 | 0.00 | 6.50 | < .0001 |
| Age | -2.18 | 1.54 | -5.26 | 0.90 | -0.12 | -1.41 | 0.162 |
| Gender (Male) | 2.24 | 1.46 | -0.68 | 5.15 | 0.13 | 1.53 | 0.130 |
| Pocket money | -0.12 | 0.15 | -0.42 | 0.17 | -0.08 | -0.82 | 0.415 |
| Years of ownership | -0.21 | 1.20 | -2.60 | 2.18 | -0.02 | -0.18 | 0.860 |
| LDR proportion | -10.47 | 6.27 | -23.00 | 2.05 | -0.15 | -1.67 | 0.100 |
| Self-control | -1.18 | 0.19 | -1.56 | -0.80 | -0.58 | -6.19 | < .0001 |
| Self-reported usage | 1.18 | 0.38 | 0.42 | 1.94 | 0.32 | 3.10 | 0.003 |

Note: Effect coding was applied for categorical variables.

## Addictive use of digital devices and academic success

Lastly, for hypothesis 3 the relationship between addictive use of digital devices and performance in the classroom was examined. The DASC scores and grade averages were not correlated (r = 0.15, p = 0.197). However, there was a positive correlation between self-reported usage of digital devices and grade average (r = 0.43, p<0.001). On average, the more time was reportedly spent with digital devices the worse the academic performance. Furthermore, self-control was also correlated to academic success (r = -0.31 p = 0.007). Again, to take into account possible effects of gender, age, years of ownership of digital devices and pocket money a regression analysis with the latter variables, self-control and self-reported usage predicting grade average was performed. As displayed in Table 4, self-control (β = -0.30, p = 0.009) and self-reported usage (β = 0.26, p = 0.040) were the only significant predictors of grade average. $R^2$ of the overall model was 0.31 with an F-statistic of 4.89 and p < 0.001. See S1 Appendix for robustness checks related to self-reported usage.

## Discussion

The main goal of this present study was to investigate the relationship between children's addictive use of digital devices and delay discounting. Consistent with previous studies on more established addictive behaviors among adolescents and adults (e.g. substance abuse, gambling, smoking), children who discounted future rewards more heavily tended to more addictively use smartphones, tablets and computers. Children showing more addictive use

**Table 4. Multiple regression analysis of predictors of grade average.**

| Term | B | SE B | 95% CI | | β | t | p |
|---|---|---|---|---|---|---|---|
| | | | LL | UL | | | |
| Intercept | 1.21 | 0.99 | -0.77 | 3.19 | 0.00 | 1.22 | 0.226 |
| Age | 0.14 | 0.08 | -0.03 | 0.30 | 0.18 | 1.68 | 0.098 |
| Gender (Male) | 0.00 | 0.08 | -0.15 | 0.15 | 0.00 | -0.02 | 0.983 |
| Pocket money | 0.01 | 0.01 | 0.00 | 0.03 | 0.21 | 1.81 | 0.075 |
| Years of ownership | 0.00 | 0.06 | -0.12 | 0.13 | 0.01 | 0.07 | 0.947 |
| Self-reported usage | 0.04 | 0.02 | 0.00 | 0.08 | 0.26 | 2.1 | 0.040 |
| Self-control | -0.02 | 0.01 | -0.04 | -0.01 | -0.30 | -2.68 | 0.009 |

Note: Effect coding was applied for categorical variables.

seem to be drawn to the immediate rewards of watching videos, gaming or social media in spite of negative long-term consequences of that behavior. The implications of this finding are twofold. First, digital addiction as a fairly new concept compares to other problematic behaviors in the context of delay discounting, suggesting its further investigation as a potentially diagnosable addiction in the future. Second, children as young as ten years old may show problematic behavior towards digital devices which has previously been observed only in adolescents and adults.

Another question addressed by this study was which role self-control played in the relationship between delay discounting and addictive use of digital devices. The regression analysis yielded that self-control confounded the relationship between the two main variables, implying that steeper discounters tended to more addictive use of digital devices due to differences in self-control. The present data suggest that children's ability to control thoughts and emotions is the mechanism underlying the association between delay discounting and addictive use and thus is a superior predictor of problematic behavior towards digital devices. Existing studies focusing on smartphone use and delay discounting found mixed results on a mediating role of self-control (mediation see Wilmer & Chein [45], no mediation see Schulz van Endert & Mohr [24]). Due to the cross-sectional and observational nature of the data, no conclusion with regard to mediation can be made in this present study [46]. Nonetheless, the moderate to strong association between self-control and addictive digital device usage found in this present study at least indicates that children who are better able to regulate thoughts and emotions tend to show a lower degree of addictive use of smartphones, tablets etc. Although the present data do not allow for firm conclusions on the direction of causality, it seems that self-controlled children resist the temptation of continued engagement with digital devices before negative effects (conflict, mood modification etc.) occur. Children lower in self-control on the other hand seem to be less able to refrain from gaming, watching videos or chatting despite recognizing adverse consequences of that behavior. Considering previous findings on the positive effect of self-control training on preventing internet addiction [47], the present finding hints at the importance of developing children's self-control in order to lower the risk of developing addictive behaviors towards digital devices.

The third association of interest was that of addictive digital device use and academic performance. Based on previous findings with other problematic behaviors, a negative relationship between these two variables was hypothesized. However, no significant association was found in this current study. Instead, self-reported usage turned out to be a significant predictor of grade average; the longer children reported to use digital devices the worse their grade average tended to be. This pattern of results suggests that children need not show symptoms of addiction, but that screen time alone may already implicate lower academic achievement. The latter finding is in line with related studies which investigated the relationship between smartphone use and students' academic success [48]. The classical interpretation for this result is that more screen time implies less study time, which leads to worse classroom performance. However, due to the correlational nature of results in this current study, the opposite causal direction cannot be ruled out. Last but not least, in line with previous large-scale studies [43, 44], self-control was found to be a significant predictor of academic success. This highlights once more the key role of children's self-control in achieving better grades already in elementary school.

The findings of this study need to be seen in light of several limitations. First, despite the (partially highly) significant results, the sample was limited in size and stemmed from one elementary school. Future studies should investigate samples from different cities and countries to allow for higher generalizability of results. Second, key variables (addictive use of digital devices, self-control, usage duration of digital devices) in this study were elicited using self-

report questionnaires. While this method is standard practice in fields such as addiction or personality research, self-estimations of screen time have been shown to diverge from actual data [49]–a phenomenon which is likely amplified due to the young age of participants. To reduce response biases, future studies might include additional sources of reports (e.g. parents, teachers or peers) or even actual screen time data, as shown in e.g. Schulz van Endert & Mohr [24]. Third, due its novelty the DASC has not yet undergone extensive validation yet. Recent research has highlighted the importance of repeated validation of psychometric scales [50, 51], which is why the results of the DASC should be interpreted tentatively at this stage. Fourth, the Monetary Choice Questionnaire is an efficient but—compared to more extensive alternatives —less sensitive instrument to assess delay discounting [52]. For example, it does not include intertemporal choices in which both rewards are given at different points in the future, which bears the risk of overweighting present bias [53]. Alternatively, adjusting delay discounting tasks, e.g. as proposed by Koffarnus & Bickel [54], could be used in future studies. Fifth, this study presented correlational results, which do not allow for causal interpretations. Looking at the reported association between self-control and addictive use of digital devices, one cannot determine using the present data whether a lack of self-control causes more addictive use or whether more problematic engagement with digital devices decreases self-control. Such conclusions may only be drawn from longitudinal or experimental studies, which are greatly needed in the future.

This study highlighted the importance of studying and monitoring the use of digital devices in children as early as at elementary school level. 10- to 13-year-olds may already show problematic behavioral patterns which have so far been only observed in older individuals. Furthermore, the central role of self-control in the context of addictive behavior as well as academic success was further underlined in this study. As experiences and influences in childhood greatly impact the trajectory of a person's entire life, researchers, politicians, educators and parents/guardians are well-advised to closely observe the impact of omnipresent digital devices on children and to assist in the development of traits which promote their well-being in the present and in the future.

## Supporting information

**S1 Table. Descriptive statistics of key variables.**
(DOCX)

**S2 Table. Correlations between DASC subscales and key variables.**
(DOCX)

**S1 Appendix. Additional correlations.**
(DOCX)

**S2 Appendix. Mediation analysis.**
(DOCX)

**S1 Dataset. Raw data from the questionnaires.**
(XLSX)

## Author Contributions

**Conceptualization:** Tim Schulz van Endert.

**Data curation:** Tim Schulz van Endert.

**Formal analysis:** Tim Schulz van Endert.

**Investigation:** Tim Schulz van Endert.

**Methodology:** Tim Schulz van Endert.

**Project administration:** Tim Schulz van Endert.

**Writing – original draft:** Tim Schulz van Endert.

**Writing – review & editing:** Tim Schulz van Endert.

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
