## [Decision Letter · Decision Letter 0]

11 May 2021

PONE-D-21-11400

Addictive use of digital devices in young children: associations with delay discounting, self-control and academic performance

PLOS ONE

Dear Dr. Schulz van Endert,

Thank you for submitting your manuscript to PLOS ONE. After careful consideration, we feel that it has merit but does not fully meet PLOS ONE’s publication criteria as it currently stands. Therefore, we invite you to submit a revised version of the manuscript that addresses the points raised during the review process.

Since the third reviewer is not satisfied with your revision, I give you the last chance to revise your manuscript. The revised manuscript will undergo the next round of review by the same reviewers. 

We look forward to receiving your revised manuscript.

Kind regards,

Baogui Xin, Ph.D.

Academic Editor

PLOS ONE

Journal Requirements:

Reviewers' comments:

Reviewer's Responses to Questions

**Comments to the Author**

1. Is the manuscript technically sound, and do the data support the conclusions?

Reviewer #1: Yes

Reviewer #2: Partly

2. Has the statistical analysis been performed appropriately and rigorously? 

Reviewer #1: Yes

Reviewer #2: Yes

3. Have the authors made all data underlying the findings in their manuscript fully available?

Reviewer #1: Yes

Reviewer #2: No

4. Is the manuscript presented in an intelligible fashion and written in standard English?

Reviewer #1: Yes

Reviewer #2: No

5. Review Comments to the Author

Reviewer #1: Having read the revised manuscript, my concerns from the initial submission have been addressed. Overall, I think that the revisions have greatly improved the manuscript to this point. At this point, I just have a few very minor comments.

Regarding the data collection, given the events of the last year, to provide important context for this study, I think it is necessary for more information on the timing of data collection to be provided. i.e., did data collection occur before any lockdowns due to the pandemic, during the time of the pandemic etc.?

For the regression analyses can you please report the R^2, F-statistic, and p-value for the overall regression in the text. This applies to all three models.

It is interesting to note that in model 1 (as described in Table 2) LDR proportion is a statistically significant predictor but, in model 2 (as described in table 3), after controlling for self-control by including it in the model, LDR proportion is no longer a statistically significant predictor of DASC scores. In model 2, the only statistically significant predictor is self-control. I’m just highlighting this as a comment, as it is already pointed out on page 13. This has important implications for the discussion as, while model 1 might seem to indicate that LDR proportion and delay discounting are important predictors of addictive digital behaviours, this appears to actually have more to do with self-control than delay discounting. When the model accounts for self-control, delay discounting is no longer a statistically significant factor predicting addictive digital behaviours. While this is briefly noted in the second paragraph of the discussion, I feel that less emphasis on the finding in the first paragraph of the discussion is warranted.

I’m curious why model 3 (as described in table 4) did not include LDR proportion and self-control as additional control variables?

Overall, these remaining concerns are minor, and I am confident that the author can address them.

Reviewer #2: I think this manuscript has addressed an increasingly common feature in children with digital devices, and how too much of a good thing can be maladaptive.

Introduction:

I appreciate the very nicely drawn up relationship between delay discounting and other addictive/maladaptive behaviours. The case for self-control is a bit sparse but acceptable. However the link to academic performance seems a bit out of context. I had an impression that the author intended to draw out a mediation / moderation analysis with these variables and possibly academic performance as dependent variable. I am a little surprised with the rather humble hypotheses. Isn't H1 and H2 essentially the same since they both have the same DV?

Method:

Clear and replicable.

Results:

The three hypotheses were simple correlations and this was further expanded into a regression.

Could the author please explain why was self-reported usage excluded from the first two analyses?

From table 2, I would run the regression as (1) DV = DASC with IV; self-reported usage, LDR proportion, self-control. They are correlated and including the non-correlated ones will remove the power of an rather small sample. (2) DV = academic performance with similar IVs.

Another analysis that I would like to see is the breakdown in the DASC. At times, the sub-scales provide a much more robust explanation to the predictors. For example, I would imagine that a child's propensity to have more conflict (as measured in the DASC) would give an idea to the power of the delay discounting and tolerance/withdrawal to self-control.

Discussion:

The author could elaborate further on the second result; with delay discounting and self-control together. What possible reasons can there be to describe this in children?

Minor comments

Method:

1. Self-control scale did not have scoring definitions i.e. higher scores mean better self-control or not?

Writing style

"A negative correlation (r=-0.28, p=0.016) between the two variables was found, implying that on average, the more often children chose the larger delayed reward, the less they addictively used digital

devices." this seems a bit convoluted and I had to re-read it a few times to understand it. I found similar writing style in other parts of results, please change them. Thanks.

6. PLOS authors have the option to publish the peer review history of their article (what does this mean?). If published, this will include your full peer review and any attached files.

Reviewer #1: No

Reviewer #2: No

---

## [Author Response · Author response to Decision Letter 0]

18 May 2021

Reviewer 1

1. Regarding the data collection, given the events of the last year, to provide important context for this study, I think it is necessary for more information on the timing of data collection to be provided. i.e., did data collection occur before any lockdowns due to the pandemic, during the time of the pandemic etc.? 

Details on the timing of data collection is now provided in lines 243-244.

2. For the regression analyses can you please report the R^2, F-statistic, and p-value for the overall regression in the text. This applies to all three models. 

The requested parameters are now included in all three models (lines 268-269, 313-314 and 531).

3. It is interesting to note that in model 1 (as described in Table 2) LDR proportion is a statistically significant predictor but, in model 2 (as described in table 3), after controlling for self-control by including it in the model, LDR proportion is no longer a statistically significant predictor of DASC scores. In model 2, the only statistically significant predictor is self-control. I’m just highlighting this as a comment, as it is already pointed out on page 13. This has important implications for the discussion as, while model 1 might seem to indicate that LDR proportion and delay discounting are important predictors of addictive digital behaviours, this appears to actually have more to do with self-control than delay discounting. When the model accounts for self-control, delay discounting is no longer a statistically significant factor predicting addictive digital behaviours. While this is briefly noted in the second paragraph of the discussion, I feel that less emphasis on the finding in the first paragraph of the discussion is warranted. 

I appreciate the reviewer’s suggestion to put more emphasis on the confounding role of self-control in the discussion. First, I have now elaborated on this finding and highlighted the superiority of self-control as a predictor of addictive use of digital devices in lines 729-732. Second, I have extended the discussion by speculating on a possibly causal mechanism behind the strong relationship between self-control and digital addiction in lines 740-745.

4. I’m curious why model 3 (as described in table 4) did not include LDR proportion and self-control as additional control variables? 

Model 3 is driven by hypothesis 3, which seeks to investigate the relationship between grade average and the DASC, the latter being replaced by self-reported usage. The control variables included in the model are standard potential confounders. LDR proportion was neither correlated with self-reported usage nor with grade average, precluding it from being a confounder in the present data. Self-control is correlated with grade average and almost significantly correlated with self-reported usage, which is why the variable is now included in model 3. Conceptually, it also makes sense to include it since it could have an influence on both self-reported usage as well as grade average.

 

Reviewer 2

1. I appreciate the very nicely drawn up relationship between delay discounting and other addictive/maladaptive behaviours. The case for self-control is a bit sparse but acceptable. However the link to academic performance seems a bit out of context. I had an impression that the author intended to draw out a mediation / moderation analysis with these variables and possibly academic performance as dependent variable. I am a little surprised with the rather humble hypotheses. Isn't H1 and H2 essentially the same since they both have the same DV? 

I appreciate the reviewer’s suggestion to enrich the cases for self-control and academic performance in the introduction. First, I added an explanation of the mechanism of self-control in the context of delay discounting in lines 126-128. Second, I provided some proposed reasons for the relationship between addictive behaviors and academic performance in lines 132-134. Third, I offered a strong rationale for studying the relationship between addictive use of digital devices and academic performance of young children in lines 137-140.

H1 and H2 are not the same, since delay discounting and self-control – despite being related – are different concepts, as described in the introduction. Therefore, they could have idiosyncratic relationships with addictive use of digital devices – which was confirmed by the present data. Also, I appreciate that “perceived significance” is not a criterion for publication in PLOS ONE.

2. Could the author please explain why was self-reported usage excluded from the first two analyses?

From table 2, I would run the regression as (1) DV = DASC with IV; self-reported usage, LDR proportion, self-control. They are correlated and including the non-correlated ones will remove the power of an rather small sample. (2) DV = academic performance with similar IVs. 

Analysis 1 (table 1) is driven by hypothesis 1, which seeks to investigate the relationship between delay discounting and the DASC. The control variables included in the model are standard potential confounders. Self-reported usage is correlated with the DASC but not with the LDR proportion, precluding it from being a confounder in the relationship between DASC and LDR proportion in the present data.

As the reviewer requested, I added self-reported usage to the second model (table 3), as it is almost significantly correlated with self-control.

Regarding model 3 (table 4), LDR proportion was neither correlated with self-reported usage nor with grade average, precluding it from being a confounder in the present data. Self-control is correlated with grade average and almost significantly correlated with self-reported usage, which is why the variable is now included in model 3. Conceptually, it also makes sense to include it since it could have an influence on both self-reported usage as well as grade average.

3. Another analysis that I would like to see is the breakdown in the DASC. At times, the sub-scales provide a much more robust explanation to the predictors. For example, I would imagine that a child's propensity to have more conflict (as measured in the DASC) would give an idea to the power of the delay discounting and tolerance/withdrawal to self-control. 

I thank the reviewer for the valuable suggestion to break down the DASC and investigate relationships with the main variables. The associations of the subscales with delay discounting are described in lines 259-262, with self-reported usage in lines 273-276 and with self-control in lines 299-301. Self-control indeed had the strongest relationship with the subscale Tolerance. The comprehensive correlation matrix is provided in S5 Table.

4. The author could elaborate further on the second result; with delay discounting and self-control together. What possible reasons can there be to describe this in children? 

I appreciate the reviewer’s suggestion to put more emphasis on the confounding role of self-control in the discussion. First, I have now elaborated on this finding and highlighted the superiority of self-control as a predictor of addictive use of digital devices in lines 729-732. Second, I have extended the discussion by speculating on a possibly causal mechanism behind the strong relationship between self-control and digital addiction in lines 740-745.

5. Self-control scale did not have scoring definitions i.e. higher scores mean better self-control or not? 

A scoring definition is now provided in lines 226-227.

6. "A negative correlation (r=-0.28, p=0.016) between the two variables was found, implying that on average, the more often children chose the larger delayed reward, the less they addictively used digital

devices." this seems a bit convoluted and I had to re-read it a few times to understand it. I found similar writing style in other parts of results, please change them. Thanks. 

The mentioned sentence was simplified in line 258. The same was done in lines 297 and 324. Any new text as part of this revision was formulated as simple as possible.

7. Have the authors made all data underlying the findings in their manuscript fully available? 

All relevant data are provided in the file “S3 Dataset”.

---

## [Decision Letter · Decision Letter 1]

28 May 2021

Addictive use of digital devices in young children: associations with delay discounting, self-control and academic performance

PONE-D-21-11400R1

Dear Dr. Schulz van Endert,

We’re pleased to inform you that your manuscript has been judged scientifically suitable for publication and will be formally accepted for publication once it meets all outstanding technical requirements.

Kind regards,

Baogui Xin, Ph.D.

Academic Editor

PLOS ONE

Additional Editor Comments (optional):

Reviewers' comments:

Reviewer's Responses to Questions

**Comments to the Author**

1. If the authors have adequately addressed your comments raised in a previous round of review and you feel that this manuscript is now acceptable for publication, you may indicate that here to bypass the “Comments to the Author” section, enter your conflict of interest statement in the “Confidential to Editor” section, and submit your "Accept" recommendation.

Reviewer #1: All comments have been addressed

2. Is the manuscript technically sound, and do the data support the conclusions?

Reviewer #1: Yes

3. Has the statistical analysis been performed appropriately and rigorously? 

Reviewer #1: Yes

4. Have the authors made all data underlying the findings in their manuscript fully available?

Reviewer #1: Yes

5. Is the manuscript presented in an intelligible fashion and written in standard English?

Reviewer #1: Yes

6. Review Comments to the Author

Reviewer #1: (No Response)

7. PLOS authors have the option to publish the peer review history of their article (what does this mean?). If published, this will include your full peer review and any attached files.

Reviewer #1: No

---

## [Editor Report · Acceptance letter]

14 Jun 2021

PONE-D-21-11400R1 

Addictive use of digital devices in young children: associations with delay discounting, self-control and academic performance 

Dear Dr. Schulz van Endert:

I'm pleased to inform you that your manuscript has been deemed suitable for publication in PLOS ONE. Congratulations! Your manuscript is now with our production department. 

Kind regards, 

on behalf of

Professor Baogui Xin 

Academic Editor

PLOS ONE